# Vaginal microbiota and personal risk factors associated with HPV status conversion—A new approach to reduce the risk of cervical cancer?

Zhongzhou Yang[1☉]*, Ye Zhang[2☉], Araceli Stubbe-Espejel[1☉], Yumei Zhao[1], Mengping Liu[3], Jianjun Li[2], Yanping Zhao[4], Guoqing Tong[5], Na Liu[1], Le Qi[1], Andrew Hutchins[6], Songqing Lin[2‡], Yantao Li[1‡]

1 BGI Genomics, BGI-Shenzhen, Shenzhen, China, 2 Department of Traditional Chinese Medicine, Women & Children Health Institute Futian Shenzhen, Shenzhen, China, 3 School of Pharmaceutical Sciences, Sun Yat-sen University, Guangzhou, China, 4 BGI-Shenzhen, Shenzhen Key Laboratory of Unknown Pathogen, Shenzhen, China, 5 Shouguang Hospital of Traditional Chinese Medicine, Reproduction Medicine Center Shanghai, China, 6 Department of Biology, Southern University of Science and Technology, Xueyuan Lu, Shenzhen, China

☉ These authors contributed equally to this work.
‡ These authors also contributed equally to this work.
* jacriyang5872@hotmail.com

**Data Availability Statement:** Both metagenomics sequence data and personal data were also deposited in the CNGB Nucleotide Sequence

## Abstract

Vaginal microbiota (VMB) is associated with changes in Human papilloma virus (HPV) status, which consequently influences the risk of cervical cancer. This association was often confounded by personal risk factors. This pilot research aimed to explore the relationship between vaginal microbiota, personal risk factors and their interactions with HPV status conversion to identify the vaginal microbiota that was associated with HPV clearance under heterogeneous personal risk factors. A total of 38 women participated by self-collecting a cervicovaginal mucus (CVM) sample that was sent for metagenomics sequencing. Most of the participants also filled in personal risk factors questionnaire through an eHealth platform and authorized the use of their previous HPV genotyping results stored in this eHealth platform. Based on the two HPV results, the participants were grouped into three cohorts, namely HPV negative, HPV persistent infection, and HPV status conversion. The relative abundance of VMB and personal factors were compared among these three cohorts. A correlation investigation was performed between VMB and the significant personal factors to characterize a robustness of the panel for HPV status change using R programming. At baseline, 12 participants were HPV-negative, and 22 were HPV-positive. Within one year, 18 women remained HPV-positive, 12 were HPV-negative and 4 participants showed HPV clearance. The factors in the eHealth questionnaire were systematically evaluated which identified several factors significantly associated with persistent HPV infection, including age, salary, history of reproductive tract infection, and the total number of sexual partners. Concurrent vaginal microbiome samples suggest that a candidate biomarker panel consisting of *Lactobacillus gasseri*, *Streptococcus agalactiae*, and *Timona prevotella* bacteria, which may be associated with HPV clearance. This pilot study indicates a stable HPV

Archive (CNSA: https://db.cngb.org/cnsa; accession number CNP0002023).

**Funding:** This study was supported by funding from the Shenzhen Innovation Committee of Science and Technology (ZDSYS20200811144002008). The funders had no role in study design, data collection and analysis, decision to publish, or preparation of the manuscript.

**Competing interests:** The authors have declared that no competing interests exist.

status-related vaginal microbe environment. To establish a robust biomarker panel for clinical use, larger cohorts will be recruited into follow-up studies.

## Introduction

An abundance of species in the vaginal microbiota (VMB) has been associated with persistent infection with high-risk human papillomavirus (HPV) and the causative agent of cervical cancer [1] as well as personal factors [2]. VMB mainly includes the larger abundances of *Lactobacillus spp*. related to HPV negativity [3]. However, HPV infection was highly relevant to protective *Lactobacillus spp*. and pathogenic *Neisseria gonorrheae*, *Chlamydia trachomatis*, *Trichomonas vaginalis*, *Mycoplasma genitalium*, *Streptococcus agalactiae* and *Timona prevotella* bacteria, which cause vaginosis [4]. Individual personal features belonging to precision medicine are beneficial to preventing persistent HPV infection or promoting HPV clearance [5].

We explored the associations between VMB and long-term HPV infection status (persistent infection or clearance) through metagenomic sequencing technology and consecutive HPV genotyping results through our digital eHealth platform. The eHealth platform was also used to collect various types of individual factors for reducing heterogeneity. Using this digital eHealth platform, our team systematically collected the personal factors that might be associated with HPV infection from the literature [6]. These factors included five categories: demographics (e.g., age), personal disease history [7], lifestyle behavior on malnutrition [8], sexual history [9–11] on the number of sexual partners and substance abuse on smoking habits [12].

To the best of our knowledge, this study was the first to identify VMB biomarkers by performing a systematic exploration of the potential confounding variables of HPV infection [2]. After obtaining metagenomics sequences and other factors through the eHealth platform, a correlation approach was utilized to explore the association between the candidate biomarkers and personal factors. We define stable microbiomes as biomarkers that are not influenced by the status of other crucial factors. The correlation p-value is utilized to select the stable biomarker panels as overlapping for each category.

## Materials and methods

### Participants recruitment and sample collection

This research was approved by the ethics committee of the Institutional Review Board at Beijing Genomics Institution (BGI-IRB 21054). This research is recorded with www.chictr.org.cn, ChiCTR2100049221. The recruitment of participants for this study began on May 25[th], 2021 and was carried out in a community setting in Shenzhen, Mainland China. Eligible participants were nonpregnant, nonlactating women who had sex at least once in their lifetime. Permission was given by the participants for the research team to use their eHealth data for both health record data including current and previous (within the last 12 months) HPV test results, as well as social personal factors. Based on the HPV test results, participants were grouped into three cohorts: HPV-negative (both samples were HPV negative), HPV-negative conversion (conversion from HPV positive to HPV negative), and HPV-positive subjects (both samples were HPV positive, suggesting persistent infection).

Once the subjects filled in their information in a registration form (S1 Text), they received a metagenomic self-sampling kit via mail, including clear instructions. Participants were requested to abstain from vaginal intercourse 24 hours before sampling, to wait for at least

three days after menstrual blood was cleared and to avoid using vaginal douches and any vaginally administered medical treatments [13–16]. A sample of vaginal mucus was collected by inserting a swab into the vagina. The swab was then stirred/placed inside a special tube with a DNA preservative solution N-octylpyridinium bromide (NOPB) [17,18]. The participants were then instructed to close the tube and place it in a plastic bag until the pick-up was arranged, and the sample was collected at room temperature for up to at least 14 days. Both the tube and the bag had a barcode/QR code for identification.

## Personal factor and eHealth platform

PROs (personal record outcomes) are defined as reports directly from the participants about the health condition status of the patient's response without interpretation or amendment by doctor or anyone else. Participants in the study were required to upload their personal PROs, which were divided into five categories with 32 personal factors (S2 Text), on the eHealth platform (CanSeq). It covers several factors, including demographics, medical history, lifestyle (S3–S6 Text), sexual history and behavior and substance abuse factors. The eHealth platform enables noninteractive support for the participants for multiple purposes. First, video and written instructions were provided on the eHealth platform to guide the participants for sample collection. Second, HPV infection records since 2016 are recorded, as authorized by users. Third, the registration of the participant´s information, including evaluated eligibility for participating in the screening program and for the collection of PROs for analytical purposes.

## Laboratory tests

After the metagenomic self-sampling kits were returned, they were sent to the China National GeneBank DataBase (CNGB). DNA was extracted for 38 samples as formerly mentioned [19–21]. Furthermore, DNA libraries were prepared as one paired-end (PE) with 350 bp insert size for individual sample [19]. A length of each read is from 75bp at stage I to 90 bp at stage II. A shotgun of metagenomic was sequenced on BGISEQ-500 platform that is equivalent with other sequencing platforms [22–24]. Data (S7 Text) analysis was carried out using an onsite pipeline, and profiles were uploaded on the online cloud pipeline [25].

The HPV test results of the participants were obtained from the eHealth platform where the SeqHPV test (BGI Shenzhen, Shenzhen, China) results of the participants were stored. The SeqHPV test is a kit to detect HPV infection in female cervical exfoliated epithelial cells by using a combinatorial probe-anchor synthesis (cPAS) sequencing approach [26]. It is utilized to detect 2 low-risk types of HPV (types: 6, 11) and 14 high-risk HPV types (types: 16, 18, 31, 33, 35, 39, 45, 51, 52, 56, 58, 59, 66, 68) [27].

## Statistical analysis

After the samples were received and sequenced, the relative abundance of the 16 VMBs was compared using *Lactobacillus* as a reference in each cohort. To differentiate the VMB profile, the relative level of microorganism abundance was applied for each cohort. The significant personal factors and microbiome were expressed as a number for categorical variables and mean ± SD for continuous variables. Analysis of variance (ANOVA) was used to compare the demographic factors. A p-value < 0.05 was considered statistically significant.

After conducting abundance and personal factor analysis, Pearson correlation analysis was applied to link the key personal indicators (S1 Table) and candidate microorganism biomarkers by linear regression for these three cohorts [28]. Student's t-test was used to determine the significance of correlation for the microorganisms in the VMB versus the HPV infection. The

resulting p-values for each microorganism were used to select candidate biomarkers for all cross-comparisons. A p-value < 0.05 was considered statistically significant for each test.

## Results

### Participant recruitment, and HPV cohorts grouping

Forty-four participants were invited to join the pilot study. Initially, 38 joined this study, but four dropped because they declined to provide their personal data (Fig 1). Thus, a total of 34 participants qualified for this study. The 34 participants completed the vaginal mucus samples and provided HPV status and personal data (S2 Table). Based on the HPV infection records within the last 12 months, the participants were grouped into three cohorts: 12 participants

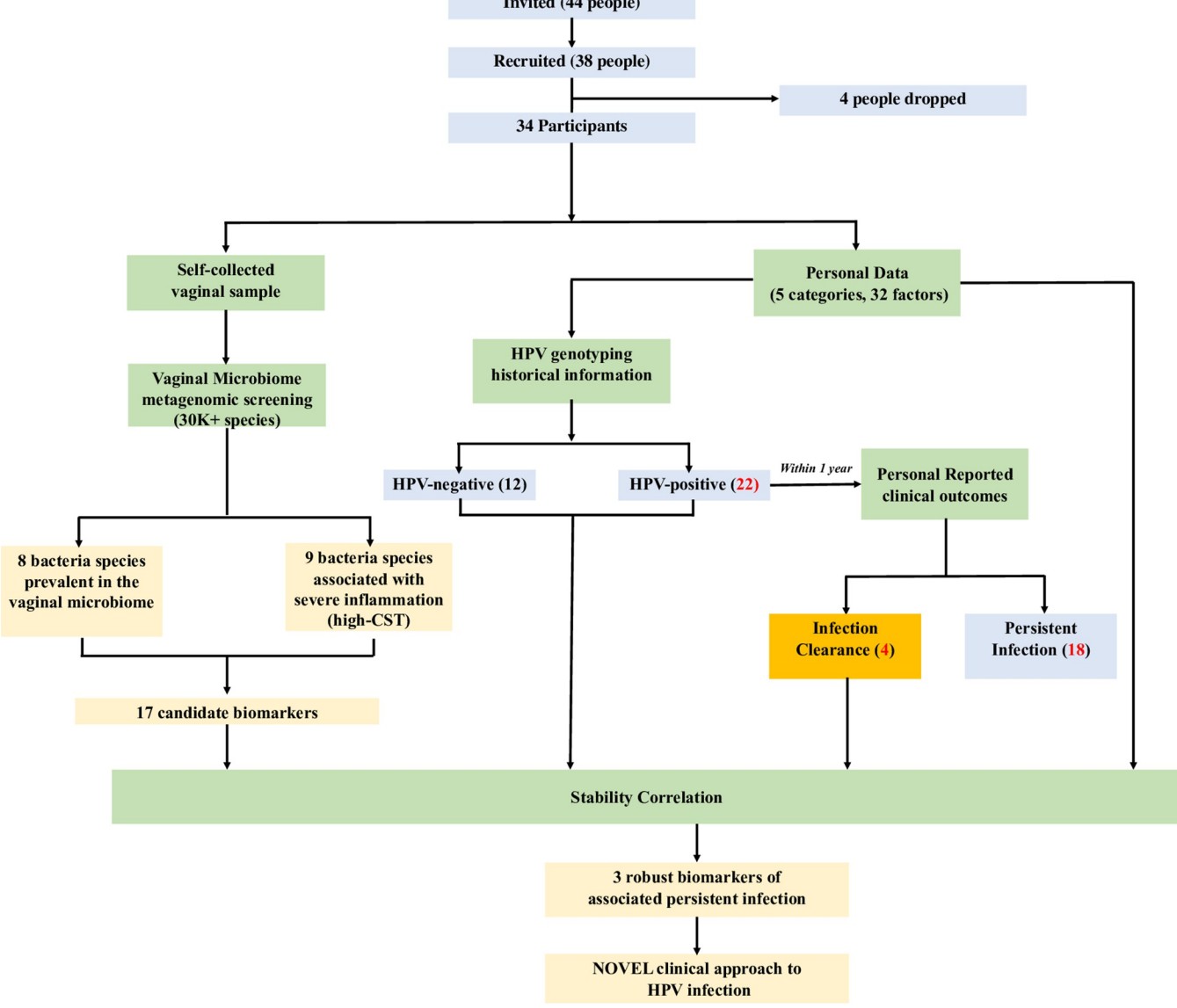

**Fig 1. Flowchart for identifying biomarkers between vaginal microbiota and HPV status.** 30K = 30,000; HPV = Human papillomavirus; CST = community state type.

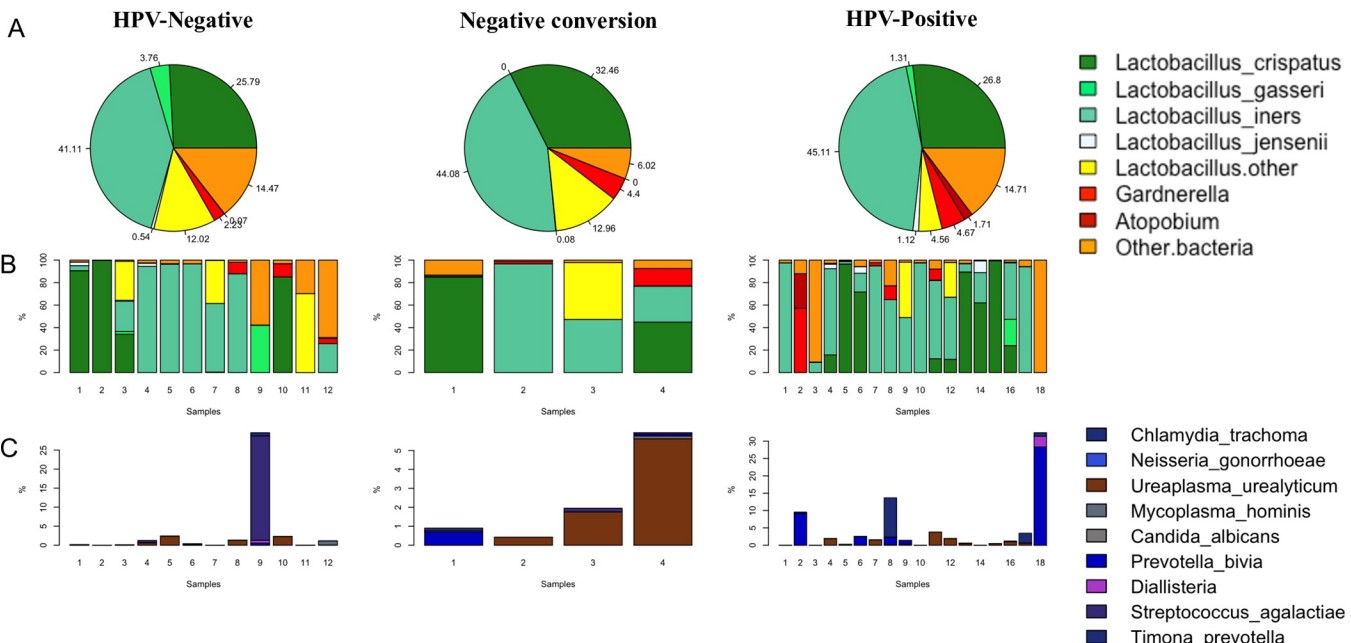

**Fig 2. Relative abundance (%) of species of microbes in the three cohorts.** Legend: **(A)** Pie charts show the relative microorganism abundance between the three cohorts. Proportion was calculated from the average value of abundance for each group by CST type. **(B)** Bar charts showing the proportion of dominant species in each sample. Selected microorganism level was selected from the CST type to show the relative abundance and characterization. **(C)** Bar charts showing the proportion of pathogenic microorganism species as indicated in the key.

were placed into the HPV-negative cohort (i.e., no HPV infection within the last year), 18 in the persistent HPV-positive cohort (persistent infection suggested by two HPV-positive results spaced 12 months apart) and 4 in the HPV positive-to-negative conversion cohort (i.e., Previously positive, but the most recent test was negative), and there were no new HPV-positive categories (i.e., Previously negative, but the most recent test was positive). Finally, we obtained 17 biomarkers to explore the relative abundance and stability correlation with personal data for five category groups. Both metagenomics sequence data and personal data were also deposited in the CNGB Nucleotide Sequence Archive (CNSA: https://db.cngb.org/cnsa; accession number CNP0002023).

## Vaginal microbiome

In the pilot stage, we focused on 17 VMBs (vaginal microbiomes), including nine community state types (CSTs) and eight gynecological diseases from the literature [29–31] through metagenomic sequencing. Overall, metagenomic sequencing identified 17 species in 9 clades (Fig 2A and 2B and S2 Text). Lactobacillus genus microorganisms were predominant in the VMB of the three cohorts, composing over 80% in most samples, agreeing closely with the patterns in previous vaginal samples [32]. In particular, *Lactobacillus iners* was identified as the predominant species among all three cohorts in this study (Fig 2A & 2B), with *Lactobacillus crispatus* being the second most abundant.

Pathogenic *Gardnerella spp* had a higher presence in HPV current or past infections. However, *atopobium* was only substantially observed in HPV-positive samples. On the other hand, all of the nine previously were identified as pathogenic gynecological infection-causing pathogens on *Trachoma chlamydia*, *Neisseria gonorrheae*, *Microureaplasma*, *Mycoplasma hominis*, *Candida albicans*, *Prevotella bivia*, *Diallisteria*, *Streptococcus agalactiae* and *Timona prevotella*.

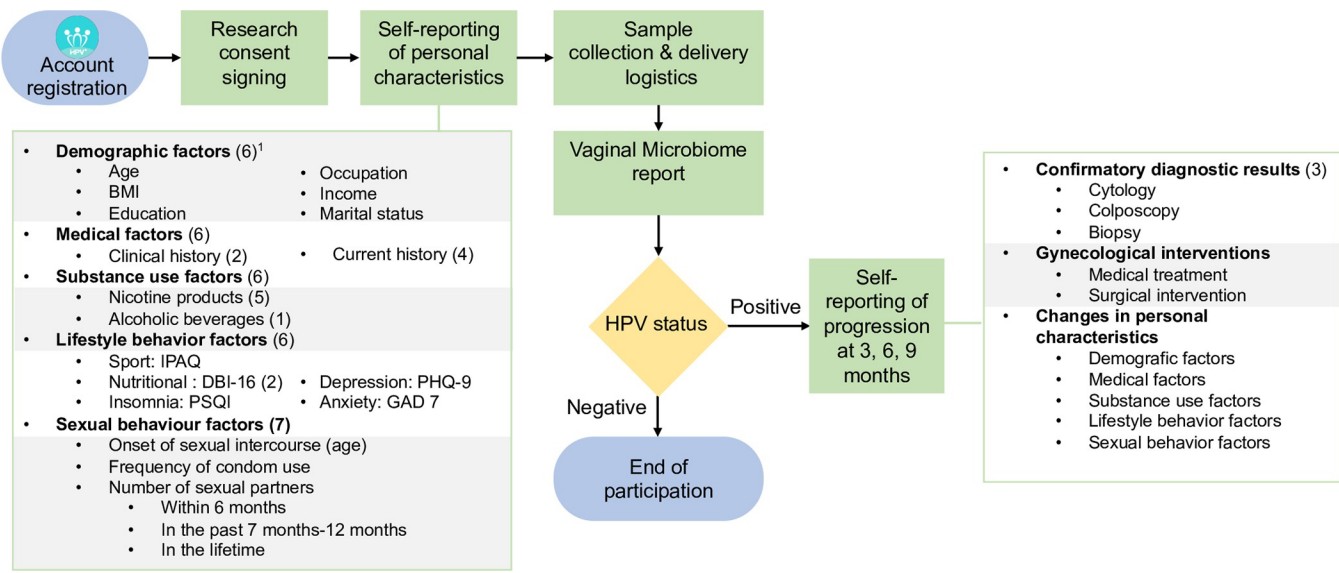

¹The number of questions is shown in the parenthesis

**Fig 3. Translational eHealth platform flowchart for the collection for participant-reported outcomes (PROs).** Legend: IPAQ = international physical activity questionnaires. DBI = diet balance index. PSQI = Pittsburgh sleep quality index. PHQ-9 = Patient Depression Questionnaire-9. GAD 7 = Generalized Anxiety Disorder 7.

They were presented in only minor proportions at 0.40% ± 2.45% among the three cohorts (Fig 2C).

## Translational eHealth platform

The eHealth application is used to interact with the participants to manage HPV test results and to collect three personal character groups and two participant-reported outcome (PRO) groups related to HPV infections (Fig 3). Participants were requested to answer a list of questions related to several factors, including simple biometrics (age, body mass index (BMI), demographic state (education, occupation, salary and marital status), medical history (six factors), substance abuse (six factors), lifestyle (six factors) and sexual history and behavior (six factors), which may affect the risk of being infected with HPV or other pathogenic microorganisms (Fig 3).

When the participants were positive for any of the HPV serotypes covered by the HPV test, they were prompted to update the eHealth questionnaires every three months. Then, the program prompts the participant to provide updates on PROs, seroconversion period, additional confirmatory diagnostic test results, and updates on their medical history, immunization (HPV vaccination) history, lifestyle changes such as starting new sports activities, changes to their usual diet, quality of sleep and psychological status, substance abuse smoking, alcohol, sexual history and behavior.

## Personal risk factors: Precision medicine

The results of the 32 PROs and their statistical association with the three types of HPV status are shown in Fig 4 (S3 Table). Demographic factors were significantly correlated with HPV status on age and salary, while education, career, BMI and matrial status were weakly correlated (Fig 4A). Fig 4B shows the medical history, including history of disease or current infection. Both reproductive tract infection (RTI) and a history of consanguineous hereditary or nonhereditary cancer seem potentially related to HPV-positive cases. Fig 4C shows the

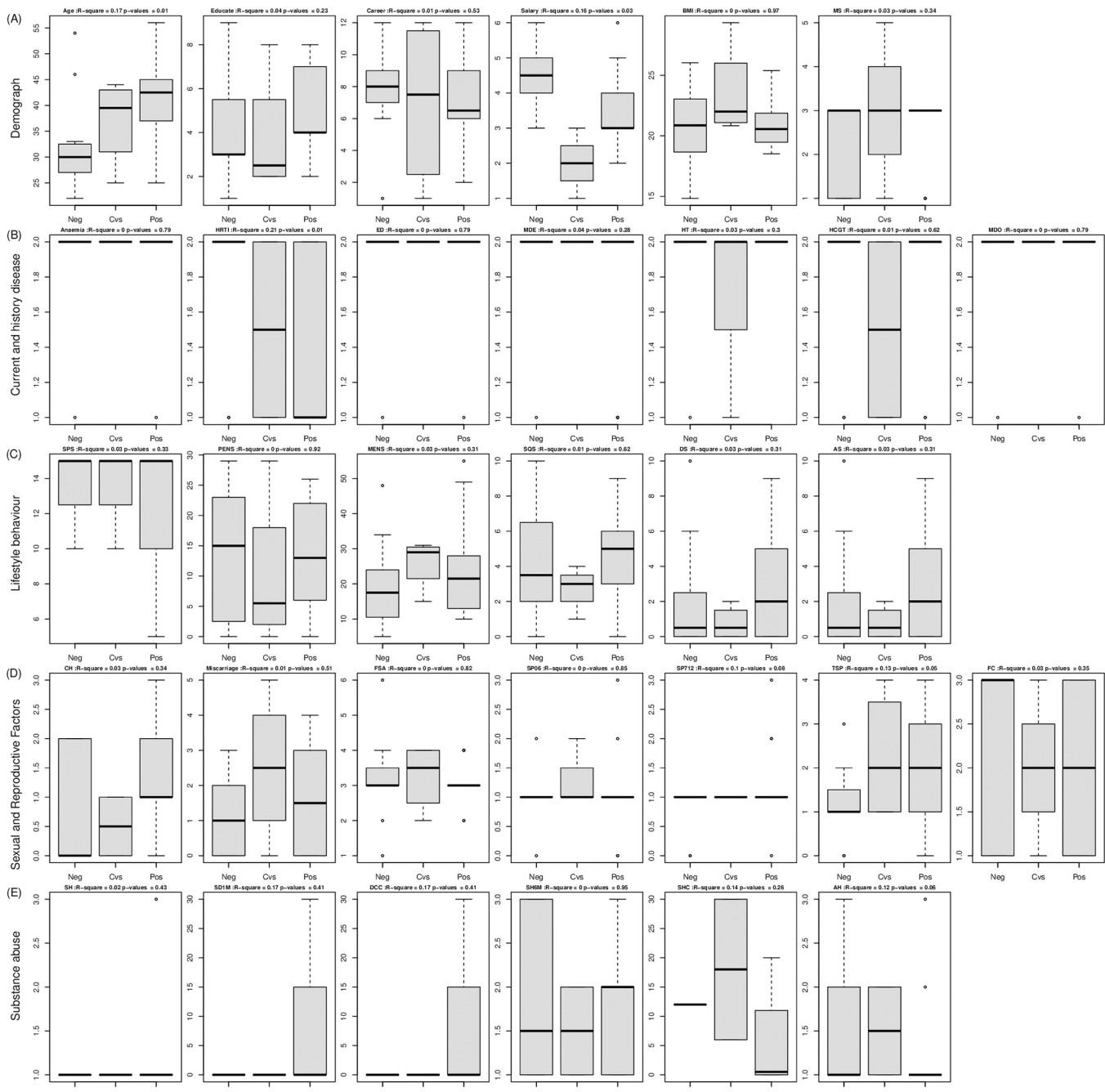

**Fig 4. Personal factors from the PROs of the participants and relation to HPV-negative, negative conversion, and HPV-positive factors.** Legend: 32 personal factors from 5 categories on three types of status. **(A)** Six demographical factors including age, educate, career and etc; MS = Marital status. **(B)** Seven medical history factors including history of disease or current infection; HRTI = History of Reproductive Tract Infection, ED = Endocrine disease, MDE = Metabolic disease, HT = History of tumor, HCGT = History of consanguineous tumor, MDO = Mental disorder. **(C)** Six behavior factors and their association with the HPV status; SPS = Sport scores, PENS = PE Nutrient scores (well nourished), MENS = ME Nutrient scores (malnourishment), SQS = Sleep quality scores, DS = Depression scores, AS = Anxiety scores. **(D)** Seven sexual and reproductive factors; CH = Childbearing history, FSA = First sexual age, SP06 = Sexual partners 0-6M, SP712 = Sexual partners 7-12M, TSP = Total sexual partners, FC = Frequency of condom. **(E)** Six substance abuse factors. SH = Smoking habit, SD1M = Smoking days within 1M, DCC = Daily cigarette consumption, SH6M = Second hand more than 6M, SHC = Secondhand cigarette, AH = Alcohol habit.

behavior factors and their association with HPV status. Malnourishment was a pseudosignificant factor for HPV infection. However, the strongest correlation between HPV status and the total number of sexual partners was also correlated (Fig 4D). However, the number of days smoking and daily cigarette consumption were not correlated with HPV status. This suggests that smoking and alcohol consumption may ultimately be indirect demographic factors (Fig 4E).

## Significant personal factors and bacteria

The baseline personal significant factors and metagenomic data are shown as eleven items in Table 1. Among the HPV-negative cohort, negative conversion and HPV-positive subjects, both age and history of reproductive tract infection had a consistent pattern. Another two significant demographic and behavioral factors were the salary range and the total number of sex partners with an inconsistent pattern. The negative HPV test results tended to be associated with higher salaries. Cohorts of participants within the lowest salary range and with the largest number of total sex partners showed greater seroconversion. HPV-negative subjects had the lowest number of total sex partners.

After adjusting for age, salary, history of reproductive tract infection and the total number of sexual partners, the metagenomics data showed that both *Lactobacillus jensenii* and *Streptococcus agalactiae* were a relatively abundant part of the VMB, and another 5 types were pseudosignificant due to the limited sample size in this pilot study. *Lactobacillus jensenii*, for example, had a relatively higher proportion in the HPV-positive group and a reduced proportion in the seroconversion group. The presence of *Streptococcus agalactiae* seemed to have a correlation between HPV-negative and HPV-positive seroconversion.

## Correlation between personal factors and microbiome

To determine the stable potential candidate biomarkers, a correlation analysis was conducted between four significant personal factors and seven microorganism species, as shown in Fig 5.

**Table 1. Significant or pseudo-significant characteristics of the participants.**

| Risk factors | HPV-Negative (12) | Negative conversion (4) | HPV-Positive (18) | p-Value [a] |
|---|---|---|---|---|
| Personal | | | | |
| Age (year) | 31.9 ± 9.3 | 37.0 ± 8.5 | 40.4 ± 8.8 | 0.01 |
| Salary[b] (¥) | 4.6 ± 1.0 | 2.0 ± 1.0 | 3.4 ± 1.1 | 0.03 |
| History of reproductive tract infection[c] | 1.8 ± 0.4 | 1.5 ± 0.6 | 1.3 ± 0.5 | 0.01 |
| Total sexual partners[d] | 1.2 ± 0.9 | 2.3 ± 1.5 | 2.1 ± 1.0 | 0.049 |
| Microorganism type | | | | |
| *Lactobacillus gasseri* | 3.8 ± 12.1 | 0 ± 0 | 1.3± 5.5 | 0.06 |
| *Lactobacillus jensenii* | 0.5 ± 1.1 | 0.1± 0.2 | 1.1 ± 2.8 | <0.01 |
| *Atopobium vaginae* | 0.1± 0.2 | 0 ± 0 | 1.7 ± 7.2 | 0.06 |
| *Mycoplasma hominis* | 0.1± 0.3 | 0 ± 0 | 0 ± 0 | 0.09 |
| *Prevotella bivia* | 0.1± 0.2 | 0.2 ± 0.3 | 2.4± 6.8 | 0.07 |
| *Streptococcus agalactiae* | 2.3± 7.9 | 0.1± 0.1 | 0 ± 0 | <0.05 |
| *Timona_prevotella* | 0.2 ± 0.3 | 0 ± 0.1 | 0.9 ± 2.7 | 0.06 |

Legend: Values are mean ± SD.

[a] Statistical difference by ANOVA (Analysis of Variance).

[b] 1: < 1000 CND; 2:1000~3000 CND; 3: 3000~5000 CND; 4: 5000~10000 CND.

[c] 1 = Yes, 2 = No, the higher the values, the lesser the probability.

[d] 3 = 3–5 partners.

[e] Adjusted for personal variables. Since this is a limited sample size, statistical difference was computed by comparing negative and positive cohorts.

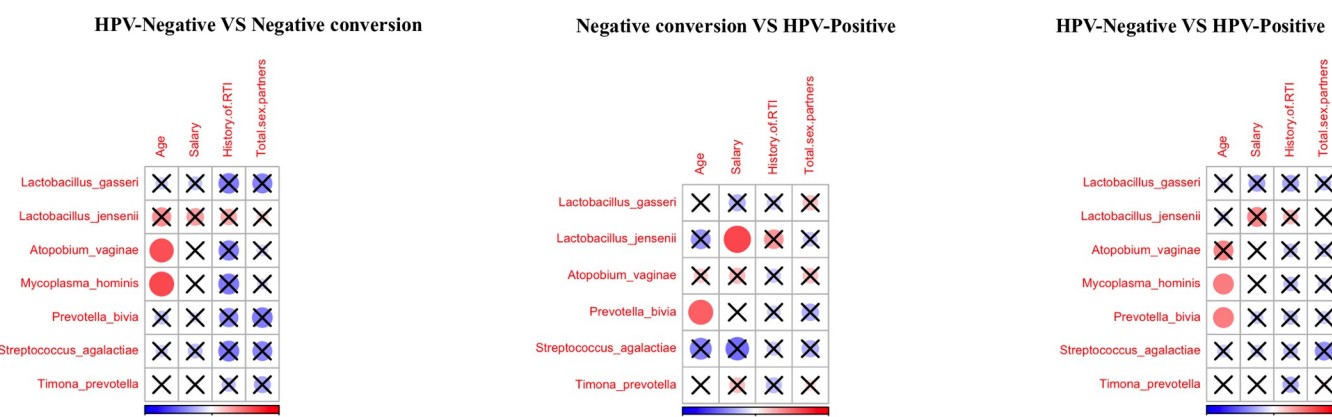

**Fig 5. Association between personal indicators the candidate biomarkers.** Legend: Correlation coefficients between four potential biomarkers and personal indicators in HPV-negative vs negative-conversion, negative-conversion vs HPV-positive, and HPV-negative vs HPV-positive cohorts. Red and blue represent positive and negative associations. Crosses represent no significant correlation (p-value > 0.05). The size of the circle represents the R-value of the personal factors and the microorganisms calculated from the linear regression.

Age has a significant association with *atopobium vaginae* and *mycoplasma hominis* in HPV-negative samples; *atopobium vaginae* and *prevotella bivia* are present in the seroconversion cases; *mycoplasma hominis* and *prevotella bivia* are abundant in the HPV-positive group. *Mycoplasma hominis* was not found in the seroconverted cohort. Other associations between personal factors and vaginal bacteria were not significant.

To identify robust biomarkers of correlation with gynecological health, finding a stable biomarker is fundamental. To increase the potential of using microbiome analysis as a useful tool in the community setting, the overlap was theoretically defined of the microorganism present within all three cohorts. *Lactobacillus gasseri*, *Streptococcus agalactiae*, and *Timona prevotella* were identified as candidate biomarkers of cervicovaginal health and differentiate HPV status.

## Discussion

This study explores the effect of the presence of different microorganisms in the vaginal microbiome of HPV-negative, HPV-positive to HPV-negative individuals and persistent HPV-positive individuals. In regard to the microorganisms that were found in vaginal mucus samples, the presence of species from the *Lactobacillus* genus dominated the microbiome, with notable representation of the *Lactobacillus iners* species. Notably, the three bacteria *Lactobacillus gasseri*, *Streptococcus agalactiae*, and *Timona prevotella* were differentially correlated to the three cohorts analyzed in this study. Overall, five microorganisms are beneficial to humans, including *Lactobacillus crispatus*, *Lactobacillus gasseri*, *Lactobacillus iners*, and *Lactobacillus jensenii*; 12 are pathogenic, including *Gardnerella vaginalis*, *Atopobium vaginae*, *Trachoma chlamydia*, *Neisseria gonorrheae*, *Microureaplasma*, *Mycoplasma hominis*, *Candida albicans*, *Prevotella bivia*, *Diallisteria*, *Streptococcus agalactiae* and *Timona prevotella*, among these 17 microorganisms. Overall, we did not find that an increased level of pathogenic bacteria was correlated with HPV status, but changes in the balance of the normal vaginal microbiome were associated with HPV infection.

Our study agrees with previous studies showing that Lactobacillus spp. are highly abundant in the vaginal microbiome [33–35]. However, the proportion of anaerobic bacteria was quite discrepant with the lower abundance. For example, *Ureaplasma urealyticum* was low at 0.55% in the HPV-negative cohort, 1.95% in the negative conversion cohort and 0.67% in the HPV-positive cohort.

We also took into consideration the 32 factors that belong to five categories, namely, demographic, medical history, lifestyle, sexual history and behavior, and substance abuse factors. Among these factors, four of them were statistically significant as age, salary, history of reproductive tract infection and total sexual partners. Specifically, history of reproductive tract infection was accounted for the association to identify biomarker panels. Other plausible factors have not accounted for the association likely, number of kids, age that women have babies, mode of delivery and HIV infection because of insignificance.

Considering the personal factors from the eHealth platform, this study first identified these three biomarkers via a correlation study. The eHealth platform is a user-friendly mobile application program that is more cost-efficient than any other kind of management and requires personal attention to every participating individual in a health-related screening program. For convenience, there is the acceptance of the Privacy Policy and research's Informed Consent. Moreover, it became the platform to report the results of the participant's clinical test, and technical assistance was provided when required. The personal feature dataset can also be used to invite the participants to enroll in additional related observational or interventional studies, such as VMB-probiotic treatment studies, VMB-screening feasibility trials and longitudinal multicenter VMB invasion research.

The burden of cervical cancer can be effectively reduced to take measures on the diagnosis, probiotic, corresponding behavior intervention and assess feasibility for future research directions. First, these three biomarkers can consist of an optimized diagnosis panel for VMB. Then, a biomarker panel can potentially be developed into probiotic bacteria for treatment. Third, the eHealth platform was potentially for lifestyle intervention, particularly for significant personal factors with the aim of minimizing the need for intervention through easy-to-follow simple instructions. One feature of a dynamic eHealth platform is that personalized feedback, a free one-to-one online medical consultation, and education training can be provided for participant-centered care. Finally, a biomarker will be evaluated in multiple centers to develop a product to reduce the risk of cervical cancer.

## Strengths and weaknesses

In this study, we have three strengths. An eHealth platform is able to gather a personal dataset to explore the stability of biomarkers. In addition, the existing metagenomic tools provide an opportunity to carry out comprehensive analyses and identify even slight variations in the abundance of microorganisms in the VMB. While 16S rRNA sequencing is traditionally used to identify the microorganism composition of the VMB, the approach is not suitable for identifying an ampler diversity of biological entities and their interrelations. Metagenomics, on the other hand, is a powerful tool that has been used to carry out broader genus searches as well as biomarker identification for drug development, as it provides one of the most compatible techniques to detect microorganisms with high reproducibility and robust reliability.

Self-sampling for vaginal mucus and vaginal epithelium cells is a relatively new feature of gynecological screening processes, which may help to remove some of the personal barriers that limit the participation of women in cervical health screening programs and expand the reach of public health interventions to remote regions by eliminating the need to have the sampling performed by a specialist in a clinical setting. Studies have shown that self-collected samples yield results that are comparable to those collected by healthcare professionals [36].

There is one major concern in this study. The sample size of this pilot study limits the statistical power of the results. The sample might not be demographically representative. To reduce the heterogeneity, an eHealth platform was utilized to deep systematically collect personal factors, which were further utilized to identify the robust biomarker panels. Additionally, our

future formal study will be the larger number of subjects to elucidate mechanisms of VMB biomarker panel.

## Conclusions

This work aimed to identify novel microorganism-based biomarkers of HPV infection, discerning among its different stages. *Metagenomic studies have shown that Lactobacillus gasseri, Streptococcus agalactiae and Timona prevotella* bacteria and their relative abundances are markedly different among different cohorts of HPV infection. An enhanced vaginal microbiota biomarker panel could be created one potential robust clinical tool for HPV acquisition, persistence or clearance, since their identification procedure is one of the biggest challenges in the ongoing colposcopy.

## Supporting information

**S1 Table. Clinical indicator.**
(XLSX)

**S2 Table. Clinical data completion.**
(XLSX)

**S3 Table. Participants clinical factor output).**
(XLSX)

**S1 Text. Recruitment procedure.**
(DOCX)

**S2 Text. Clinical factors by questionnaire.**
(DOCX)

**S3 Text. Personal factors on sport (IPAQ).**
(PDF)

**S4 Text. Personal factors on insomnia (PSQI).**
(PDF)

**S5 Text. Personal factors on Depression (PHQ-9).**
(PDF)

**S6 Text. Personal factors on Anxiety (GAD-7).**
(PDF)

**S7 Text. Metagenomics.**
(DOCX)

## Acknowledgments

We acknowledge the BGI IT Team (Chungen Zhang, Hao Zhou and Weiwei Bai) for developing the eHealth platform and Kaiye Cai for her bioinformatics assistance.

## Author Contributions

**Conceptualization:** Zhongzhou Yang, Ye Zhang, Yantao Li.

**Data curation:** Araceli Stubbe-Espejel, Yumei Zhao, Jianjun Li, Yantao Li.

**Formal analysis:** Zhongzhou Yang, Mengping Liu, Yanping Zhao, Andrew Hutchins.

**Funding acquisition:** Yantao Li.

**Investigation:** Zhongzhou Yang, Araceli Stubbe-Espejel, Yanping Zhao, Guoqing Tong.

**Methodology:** Zhongzhou Yang, Araceli Stubbe-Espejel, Mengping Liu.

**Resources:** Yumei Zhao, Jianjun Li, Na Liu, Le Qi, Yantao Li.

**Software:** Zhongzhou Yang, Le Qi, Songqing Lin.

**Visualization:** Zhongzhou Yang.

**Writing – original draft:** Zhongzhou Yang.

**Writing – review & editing:** Zhongzhou Yang, Araceli Stubbe-Espejel, Yanping Zhao, Andrew Hutchins.

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
