## [Decision Letter · Decision Letter 0]

2 Mar 2022

PONE-D-21-39382Vaginal microbiota and personal risk factors associated with HPV status conversion – a new approach to reduce the risk of cervical cancer?PLOS ONE

Dear Dr. Yang,

Thank you for submitting your manuscript to PLOS ONE. After careful consideration, we feel that it has merit but does not fully meet PLOS ONE’s publication criteria as it currently stands. Therefore, we invite you to submit a revised version of the manuscript that addresses the points raised during the review process.

We look forward to receiving your revised manuscript.

Kind regards,

Guangming Zhong

Academic Editor

PLOS ONE

Journal Requirements:

"No"

"No"

 This information should be included in your cover letter; we will change the online submission form on your behalf

Reviewers' comments:

Reviewer's Responses to Questions

**Comments to the Author**

1. Is the manuscript technically sound, and do the data support the conclusions?

Reviewer #1: Yes

2. Has the statistical analysis been performed appropriately and rigorously? 

Reviewer #1: Yes

3. Have the authors made all data underlying the findings in their manuscript fully available?

Reviewer #1: Yes

4. Is the manuscript presented in an intelligible fashion and written in standard English?

Reviewer #1: Yes

5. Review Comments to the Author

Reviewer #1: Human papillomavirus (HPV), being considered as a sexually transmitted pathogen, which is responsible for over 90% of cervical cancer cases, poses a severe threat to woman’s reproductive health. Though the factors affecting HPV-persistence are not fully understood, emerging data suggests that there exists an association between cervical HPV infections and the vaginal microbiota. This study demonstrated that several factors significantly associated with persistent HPV infection, including age, salary, history of reproductive tract infection, and the total number of sexual partners and in vaginal microbiota, Lactobacillus gasseri, Streptococcus agalactiae, and Timona prevotella bacteria may be associated with HPV clearance. Besides, by using the eHealth platform which is a user-friendly mobile application program, it is more cost-efficient than any other kind of management for researchers to pay full attention to every participant in a health-related screening program.

1. Major

Line 101 to line 104:

We found the participant was requested to abstain from vaginal intercourse 24 hours before sampling, to wait for at least three days after menstrual blood was cleared and to avoid using vaginal douches and any vaginally administered medical treatments. But at different stages of the menstrual cycle, such as follicular phase and the luteal phase, physiological changes could have an impact on vaginal flora growth, colonization, and community structure. It is suggested to collect sample of vaginal mucus at the same stage of the menstrual cycle or indicate what stage of the menstrual cycle the collected samples come from.

2. Minor

1) Line 105 to line 106:

Sample of vaginal mucus was collected by inserting a swab into the vagina, which was then stirred/placed inside a special tube with a preservative solution at room temperature until the pick-up was arranged. Is this preservative solution a DNA protection solution?

2) Line 277:

“HPV-negative to HPV-negative” would be “HPV positive to HPV negative”.

6. PLOS authors have the option to publish the peer review history of their article (what does this mean?). If published, this will include your full peer review and any attached files.

Reviewer #1: No

---

## [Author Response · Author response to Decision Letter 0]

26 Mar 2022

Dear Reviewer:

Sincerely thanks for your three comments. They are responding in below:

1. Major

Line 101 to line 104:

We found the participant was requested to abstain from vaginal intercourse 24 hours before sampling, to wait for at least three days after menstrual blood was cleared and to avoid using vaginal douches and any vaginally administered medical treatments. But at different stages of the menstrual cycle, such as follicular phase and the luteal phase, physiological changes could have an impact on vaginal flora growth, colonization, and community structure. It is suggested to collect sample of vaginal mucus at the same stage of the menstrual cycle or indicate what stage of the menstrual cycle the collected samples come from.

Thanks for your questions. We collected samples after 3 days beyond the menses. It can be both follicular and luteal phase because of three reasons with four references as in manuscript.

1.1, According to Stephanie et al, vaginal microbial diversity, as measured using the Shannon index, increased during menses blood (P < 0.001), while Lactobacillus abundances decreased (P = 0.01). Hence, we did not collect samples during menses. 

1.2, Based on Bonnie et al, the overall vaginal microbiome of most women remained relatively stable throughout the menstrual cycle, with little variation in diversity and only modest fluctuations in species richness. That is to say, there is little variation between follicular and luteal phase. 

1.3, The reason why it was three days accounted by Pawel et al. In their article, Figure S6 showed Shannon diversity indices over the menstrual time. Based on the figure, the diversity index decreased to one and less after three days in the end of menses. 

1.4, Although samples obtained during a menstrual period should be valid, most women would prefer to obtain the sample at a time other than during their menstrual flow based on Jerome et al. 

Therefore, we summarized the descriptions for: to wait for at least three days after menstrual blood was cleared.

2. Minor

1) Line 105 to line 106:

Sample of vaginal mucus was collected by inserting a swab into the vagina, which was then stirred/placed inside a special tube with a preservative solution at room temperature until the pick-up was arranged. Is this preservative solution a DNA protection solution?

Thanks for your questions. Exactly, this solution is a DNA protection solution through N-octylpyridinium bromide (NOPB) published in Han-Microbiome-2018 from my institution. Furthermore, the NOPB was also acknowledged by article Qian-Chin Med J (Engl)-2020. Two articles are shown in manuscripts 

2) Line 277:

“HPV-negative to HPV-negative” would be “HPV positive to HPV negative”.

Thanks for your suggestions. It was updated in accordingly. 

Dear Editor:

With regard to your three additional requirements, they were responded in below:

Responses: To easy review for you, files have been updated carefully to ensure meets PLOS ONE's style requirements after reading your suggested documents. 

2. Stating the financial disclosure:

Responses: This study was supported by funding from the Shenzhen Innovation Committee of Science and Technology (ZDSYS20200811144002008). The funders had no role in study design, data collection and analysis, decision to publish, or preparation of the manuscript.

3. Stating the competing interests section: 

Responses: The authors have declared that no competing interests exist. 

Jacri

---

## [Editor Report · Decision Letter 1]

12 Jun 2022

Vaginal microbiota and personal risk factors associated with HPV status conversion – a new approach to reduce the risk of cervical cancer?

PONE-D-21-39382R1

Dear Dr. Yang,

We’re pleased to inform you that your manuscript has been judged scientifically suitable for publication and will be formally accepted for publication once it meets all outstanding technical requirements.

Kind regards,

Guangming Zhong

Academic Editor

PLOS ONE
---

## [Editor Report · Acceptance letter]

29 Jul 2022

PONE-D-21-39382R1 

Vaginal microbiota and personal risk factors associated with HPV status conversion – a new approach to reduce the risk of cervical cancer? 

Dear Dr. Yang:

I'm pleased to inform you that your manuscript has been deemed suitable for publication in PLOS ONE. Congratulations! Your manuscript is now with our production department. 

Kind regards, 

on behalf of

Dr. Guangming Zhong 

Academic Editor

PLOS ONE